# Layout Transposition for Non-Visual Navigation of Web Pages by Tactile Feedback on Mobile Devices

**DOI:** 10.3390/mi11040376

**Published:** 2020-04-03

**Authors:** Fabrice Maurel, Gaël Dias, Waseem Safi, Jean-Marc Routoure, Pierre Beust

**Affiliations:** 1Groupe de Recherche en Informatique, Automatique, Image et Instrumentation (GREYC), National Graduate School of Engineering and Research Center (ENSICAEN), Université de Caen Normandie (UNICAEN), 14000 Caen, France; gael.dias@unicaen.fr (G.D.); jean-marc.routoure@unicaen.fr (J.-M.R.); pierre.beust@unicaen.fr (P.B.); 2Higher Institute for Applied Sciences and Technology, Damascus, Syria

**Keywords:** vibrotactile feedback, blind users, web accessibility

## Abstract

In this paper, we present the results of an empirical study that aims to evaluate the performance of sighted and blind people to discriminate web page structures using vibrotactile feedback. The proposed visuo-tactile substitution system is based on a portable and economical solution that can be used in noisy and public environments. It converts the visual structures of web pages into tactile landscapes that can be explored on any mobile touchscreen device. The light contrasts overflown by the fingers are dynamically captured, sent to a micro-controller, translated into vibrating patterns that vary in intensity, frequency and temperature, and then reproduced by our actuators on the skin at the location defined by the user. The performance of the proposed system is measured in terms of perception of frequency and intensity thresholds and qualitative understanding of the shapes displayed.

## 1. Introduction

Voice synthesis and Braille devices are the main technologies used by screen readers to afford access to information for visually impaired people (VIP). However, they remain ineffective under certain environmental conditions, in particular on mobile supports. Indeed, in 2017, the *World Health Organization* (WHO: https://www.who.int/) counted about 45 million blind people in the world. However, a study by *The American Printing House for the Blind* (APH: https://www.aph.org/) showed that less than 10% of children between the ages of 4 and 21 are Braille readers. This statistic is even lower for elder populations. Therefore, improving access to the Web is a priority, particularly to promote the autonomy of VIP, who do not practice Braille.

At the same time, Web information is characterized by a multi-application multi-task multi-object logic that builds complex visual structures. As such, typographical and layout properties used by web designers allow sighted users to take a large amount of information in a matter of seconds to activate appropriate reading strategies (first glance view of a web page). However, screen readers, which use voice synthesis struggle to offer equivalent non-linear reading capabilities in non-visual environments. Indeed, most accessibility software embedded in tablets and smartphones synthesize the text orally as it is overflown by the fingers. This solution is interesting but daunting when it comes to browsing new documents. In this case, the blind user must first interpret and relate snippets of speech synthesis to the organization of all page elements. Indeed, the interpretation of a web page can only be complete if the overall structure is accessible (aka. morpho-dispositional semantics). To do this, he moves his fingers over almost the entire screen to carry out a heavy and somewhat random learning phase (often too incomplete to bring out rapid reading strategies). In fact, most users rarely do so and have a “utilitarian” practice of touch devices, confining themselves to the functionalities of the web sites and interfaces they are perfectly familiar with.

To reduce the digital divide and promote the right to “stroll” to everyone, it is imperative to allow the non-visual understanding of the informational and organizational structures of web pages. When a sighted person reads a document silently, we often observe a sequence of specific and largely automated mental micro-processes: (1) the reader takes information from a first glance of all or part of the web page by an instantaneous overview (skimming); (2) he can also activate fast scans of the medium (scanning) consisting of searching for specific information within the web page. These two processes, which are more or less conscious, alternate local and global perception, and can be repeated in different combinations until individual objectives are met. Then, they can be anchored in high-level reading strategies, depending on reading constraints and intentions. As such, layout and typography play a decisive role in the success and efficiency of these processes. However, their restitution is almost absent in existing screen reader solutions.

The purpose of our research is to provide access to the visual structure of a web page in a non-visual environment, so that the morpho-dispositional semantics can be accessed by VIP and consequently enable the complete access to the informative message conveyed in a web page. For that purpose, we propose to develop a vibrotactile device (called TactiNET), which converts the visual structure of a web page into tactile landscapes that can be explored on any mobile touchscreen device. The light contrasts overflown by the fingers are dynamically captured, sent to a micro-controller, translated into vibrating patterns that may vary in intensity, frequency and temperature, and then reproduced by our actuators on the skin at the location defined by the user. In this paper, we particularly focus on the skimming strategy and leave for further work the study of scanning procedures. Consequently, we specifically tackle two research questions that are enunciated below:**Question 1:** What are the frequency and intensity thresholds of the device such that maximum perception capabilities can be obtained (the study of temperature is out of the scope of this paper)?**Question 2:** How efficient is the device to provide access and qualitative understanding of the visual shapes displayed in a web page?

The paper is organized in 6 sections. In the next section, we detail the related work. In Section 3, we provide all the technical issues of the TactiNET. In Section 4, we present the results of the study of the sensory capabilities of the device. In Section 5, we describe the experiments conducted to evaluate the efficiency of the device to reproduce the visual structure of a web page. Finally, in Section 6, we provide some discussions and draw our conclusions.

## 2. Related Work

Numerous devices have been proposed to attach vibrotactile actuators to the users’ body to increase the perception and memorization of information. The idea of a dynamic sensory substitution system can be found as early as the 1920s as mentioned in [1]. In the specific case of the transposition of visual information into the form of tactile stimuli, a series of remarkable early experiments are described in [2], which coined the term Tactile Vision Sensory Substitution (TVSS) for this purpose. In this case, 400 solenoid actuators are integrated into the backrest of a chair and the user seated on it manipulates a camera to scan objects placed in front of him. The images captured are then translated into vibratory stimuli that are dynamically transmitted to the actuators. The spectacular results from these experiments demonstrate the power of human brain plasticity to (1) naturally exploit visual information encoded in a substitute sensory modality, (2) externalize its sensations and (3) create meaning in a manner comparable to that which would have been produced by visual perception.

A few years later, the Optacon has been proposed to offer vibrotactile feedback [3]. This particularly innovative device is capable of transposing printed texts into vibrotactile stimuli. An optical stylus and a ruler make it possible to follow the lines of a text and to reproduce the shapes of the letters dynamically under the pulp of a finger positioned on vibrotactile pins. Three weeks of learning on average were sufficient for a reading performance of about 16 words per minute, which stands for a remarkable result as it opened up the new possibility of accessing non-specific paper books, i.e., designed for sighted people. The interesting idea is to support sensory substitution through active exploration and analog rather than symbolic transposition (compared to Braille, for example in the D.E.L.T.A system [4]). Unfortunately, despite its very good reception at the time by the blind population, the marketing of the product was short due to the lack of a sustainable economic balance.

More recently, with the advent of new portable technologies and the constant increase in the power of embedded applications and actuators, we can observe many interesting studies for the use of touch in new interactions. [5] brings interesting knowledge about the potential of rich tactile notifications on smartphones with different locations for actuators, and intensity and temporal variations of vibration. Ref. [6] present a simple and inexpensive device that incorporates dynamic and interactive haptics into tabletop interaction. In particular, they focus on how a person may interact with the friction, height, texture and malleability of digital objects. Other research devices exploiting the ideas developed in TVSS are dedicated to improving the perception of blind people to facilitate their independent movements. To this end, Ref [7] propose to automatically extract the contours of images captured in the blind person’s environment. This piece of information is then transposed into vibratory stimuli by a set of 48 motors integrated inside a vest worn by the subject. As such, this navigation system (Tyflos) integrates a 2D vibration array, which offers to the blind user a sensation of the 3D surrounding space.

In a task-oriented approach, these proposals do not adequately cover our needs regarding the consideration of typography and layout for the non-visual reading of web pages. Several more specific studies come closer to our perspectives. Many studies have focused on the use of textures to produce different tactile sensations during the spatial exploration of graphics [8]. This research has led to recommendations on texture composition parameters in terms of elementary shape, size, density, spacing, combination or orientation. From there, devices have been developed to facilitate access to diagrams by blind people [9,10,11,12,13,14]. However, they do not tackle the overall complexity of multi-modal web documents that may gather textual, visual, layout information, to name but a few.

To fulfill this gap, the first tactile web browser adapted to hypertext documents was proposed by [15]. Filters were applied to images and graphics to reduce the density of visual information and extract important information to be displayed on a dedicated touch screen. The text was rendered on the same touch screen in 8-dot Braille coding. This browser illustrates three main limitations we wish to remove. First, it only considers part of the layout information (layout of the elements, graphic/image/text differences), which are not sufficient to exploit the richness of the typographical phenomena (e.g., colors, weights, font size) and the luminous contrasts they induce. Second, the browser uses Braille to render text on the screen, whereas only a minority of blind people can read it and the Braille language is limited to translate typographic and layout information. Third, the user’s autonomy is reduced in the choice of accessible information since the browser unilaterally decides which information is likely to be of interest. Another interesting browser has been proposed called CSurf [16] but it also relies on data filtering and the valuable information is selected by the browser itself. TactoWeb [17] is a multi-modal web browser that allows visually impaired people to navigate through the space of web pages thanks to tactile and audio feedback. It relies on a lateral device to provide tactile feedback consisting of a cell that stimulates the fingertip by stretching and contracting the skin laterally. Although it preserves the positions and dimensions of the HTML elements, TactoWeb sorts and adapts the information based on the analysis of the structure of the Document Object Model (DOM) tree. Closer to the idea of the Tactos system [9] applied to web pages, the browser proposed by [18] requires a tactile mouse to communicate the presence of HTML patterns hovering over the cursor. The mouse is equipped with two touch cells positioned on top. During the system evaluations, web pages were presented to the participants and then explored using the device. Each blind user was asked to describe the layout of the visited pages. The results indicate that while the overall layout seemed to be perceived, the description still revealed some inconsistencies in the relationship between the elements and in the perception of the size of the objects. The idea of an analogical tactile perception of a web page is appealing but only if the tactile vocabulary of the device is rich enough to transpose the visual semantics of web pages. Moreover, the disadvantage of requiring a specific web browser partially breaks the principle, which we call “design for more”, on which we wish to base our solution. This approach comes from the observation that one of the reasons that weakens the appropriation of a device by a blind person is its destructive aspect. We make the hypothesis that a tool, even if it offers new useful features, may not be accepted if it prevents the use of widely tested features. The system should be able to be added to and combined with the tools classically used by a given individual, whether with a speech synthesis, a Braille track or a specific browser.

According to us, a triple objective must be guaranteed to develop successful TVSS devices in the context of non-visual access of web information: perception, action and emotion. Indeed, the typographical and layout choices contribute to giving texture and color to the perception of the document. As such, the author transmits a certain emotional value using these contrasts. Therefore, we make the hypothesis that the improvement of screen readers goes through the development of devices that make it possible to perceive the coherence of the visual structure of documents, as much as for the information it contains, and the interactions it suggests, and the emotional value it conveys.

Another important aspect for the success of powerful screen readers consists of guaranteeing conditions of appropriation. First, the system must not hinder exploratory movements, be autonomous, robust and light. It must also be discrete and inexpensive as well as be easy to remove. The device must also meet real user needs. Indeed, the multiplication of digital reading devices greatly complicates the visual structure of digital documents in general and web pages in particular. There is therefore a great need for this population to facilitate non-visual navigation on the Internet. We designed the TactiNET device, presented in the following subsection, to meet this demand.

## 3. TactiNET Hardware and Framework

The state of the art shows that interfaces for non-visual and non-linear access to web pages are still limited, especially with nomadic media, i.e., the blind user perceives the document only through fragments ordered in the temporal dimension alone. It is imperative to allow a non-visual apprehension of the documents that is both global and naturally interactive by giving blind people a tactile representation of the visual structures. Our ambition is to replace one’s capacity for visual exploration, which relies on the luminous vibration of the screen, by a capacity for manual exploration, which relies on the tactile perception of vibrotactile or thermal (or both) actuators.

The device should translate information as faithfully as possible while preserving both informational and cognitive efficiency. The user will then be in charge of interpreting the perceived elements and will be able to make decisions that will facilitate active discovery, learning and even bypassing the initially intended uses (concept known as enactivism [19]). This idea runs counter to most of the current attempts to produce intelligent technologies by building complex applications whose Man-Machine coupling is thought upstream. In this objective, a metaphor, known as “white cane metaphor”, guides both our hardware and software design, i.e., the blind person explores the world by navigating thanks to the contacts of his cane with the obstacles and materials around him. We hope that the semantics of the text visual structures will play this role for the digital exploration of documents, by creating a sensory environment made of “text sidewalks”, graphical textures and naturally signposted paths orienting the movements of this “finger-cane”. The TactiNET hardware [20] has been developed to offer both versatility and easy setup in the design of experiments. As shown in Figure 1, it consists of:A control tablet (item **1**) where all the experimental parameters can be managed and programmed (e.g., shape of the patterns, vibrations frequencies, etc.) and sent via Bluetooth to the user’s tablet (item **2**).A user tablet (item **2**), where web pages are processed and displayed according to the graphical language. The user’s five fingers coordinates are sent to the host system via a connection provided by a ZigBee dongle (item **3**).The host system (item **4**) where actuators can be connected through satellites. Each actuator can be either a piezoelectric vibrator or a Peltier device to provide haptic or thermal user feedback (item **5**).

Depending on the information flown over by the fingers, data packets containing the control information are transmitted from the user tablet to the host system, which can then control the piezoelectric vibrators and Peltier based on the requested amplitude, frequency and temperature values. All this hardware setup has been designed and realized at the GREYC UMR 6072 (http://www.greyc.fr) laboratory with plastic cases built with a 3D printer. A total control on all the experimental parameters is thus achieved and future extensions can be easily implemented. The host system is battery powered with a built-in USB battery management system. It is thus portable and provides more than 2 h of experiment time with full charge.

### 3.1. Host System and Actuators Descriptions

The host system is built based on an Atmel ATZ-24-A2R processor that communicates with the user’s tablet with its built-in ZigBee dedicated circuits. The host system consists of one main board with processor, battery management and communication circuits, and up to 8 daughter boards that can be stacked. Each daughter board can control up to 4 independent actuators connected via standard 3.5 mm 4 points connectors using an I2C protocol (see the satellite in Figure 2). Two kinds of actuators have been developed to provide both haptic and thermal feedback: piezoelectric and Peltier.

First, classical haptic devices usually use unbalanced mass small electric motors. In such devices, the intensity and the frequency of the vibration are completely correlated. To avoid this correlation and to have a perfect control on these two important parameters for haptic feedback, a more sophisticated actuator based on the piezoelectric effect has been used. In particular, a specific circuit from Texas Instrument was used to generate the high voltage (up to 200 V) needed to control the actuator.

Second, a simple thermal effect can be achieved using a power resistor. In such a device, only temperature increase can be generated. To have a better control on the thermal feedback, a Peltier module has been used. As such, by controlling the sign of the DC voltage, the user’s skin can be cooled or heated. The DC voltage is generated with a dedicated H bridge circuit.

Each satellite board contains the actuators and the dedicated circuits needed to control it. The type of each satellite can be identified by the host system. This hardware architecture provides versatility and is easy to use. As an example, Figure 3 shows a configuration with the host system wrapped around the wrist with three satellites providing two kinds of haptic feedbacks and a thermal one.

### 3.2. Performances

The haptic feedback consists of vibrations in a frequency range from 50 Hz up to 550 Hz with a resolution of 7 Hz. The intensity of the vibration should be given in terms of skin pressure. Unfortunately, one should know the mechanical force applied by the actuators on the skin. This parameter is very difficult to obtain since it depends on the skin resistance that varies a lot from user to user and with the environment (temperature, relative humidity, etc.). As we will see in the next section, the vibration intensities are thus given by the 8 bit number used in our protocol to control the vibration (0 = no vibration and 255 = full vibration).

The thermal feedback consists of temperature variations limited to +/−5 °C: the main limitation is due to the DC current needed that may reduce the experiment time. Up to +/−10 °C can be easily achieved but with serious time limitations.

With the ZigBee protocol, up to 10 host devices can be addressed. In addition, each host device can receive 8 daughter boards that can control 4 actuators. Finally, up to 320 actuators can be controlled within the TactiNET.

In summary, the device is designed as a modular experimental prototype intended for researchers who are not experts in electronics. As such, depending on the objective of the study, different combinations can easily be composed and evaluated, both in terms of the number of actuators and their type. In addition, the actuators can freely be exchanged in a plug-and-play mode and are integrated in a plastic housing made by a 3D printer. Each element has a hooking system so that it can be positioned on different parts of the user’s body. In our experiments, described in the following subsections, a single satellite is used with haptic feedback and placed on the non-navigating hand.

## 4. The Graphical Language of the TactiNET

In this section, we first propose an empirical validation of the TactiNET framework to recognize simple shapes that simulate web page layouts, and based on demonstrated limitations, we define the foundations of a graphical language based on patterns that vary in intensity and frequency.

### 4.1. Towards a Graphical Language for the Tactile Output

A first experiment has been carried out in [21] that aims at pre-testing the ability of users to recognize shapes on a handheld device with the TactiNET in its minimal configuration within a non-visual environment. This configuration consists of:A single satellite positioned on the non-active hand: one vibrotactile actuator,A single dimension of variation: the brighter (respectively darker) the pixel overflown by the index finger, the lower (respectively higher) the vibration amplitude.

The experiment has been conducted with 15 sighted users (eyes closed) and 5 blind people (see Table 1) to explore, count, recognize and manually redraw different simulated web pages configurations. The conclusions of this experiment were as follows. First, the ability to distinguish the size of the shapes and their spatial relationships was assessed as highly variable in terms of exploration time (7 to 20 min in total to explore 4 configurations). The quality of manual drawings also varied from very bad to almost perfect depending on user characteristics (age, early blindness, familiarity with tactile technologies).

However, as demonstrated in Figure 4, the best productions are qualitatively interesting despite the relatively simple configuration of the TactiNET. Other interesting side results are worth noting. First, an encouraging learning effect was clearly demonstrated when the experience lasted no more than an hour. Second, we have identified a metric to measure a user’s interest in the information overflown: the greater the interest, the proportionally greater the pressure exerted on the screen (this feature needs to be studied more deeply in future work, in particular for scanning purposes).

To propose a more complete graphical language capable of handling more relationships between (1) the layout and typographic clues, and (2) the graphical patterns variable in shapes, sizes, surface and border textures, and distances, we proposed to improve the perceptive capacities of the TactiNET by optimizing the combination of different tactile stimuli, namely amplitude and frequency.

### 4.2. Minimum Perception Thresholds of Frequency and Amplitude

From this perspective, we designed a second experiment [22] designed to select the most perceptible frequency range of the TactiNET device. Each frequency value studied was further combined with an amplitude value either constant or increased by a slight random variability in order to incorporate a texture effect that may enable more perceptual capacities. For that purpose, we conducted an experiment that crossed 3 groups of users (38 sighted children, 25 sighted adults and 5 blind adults, the same population as in Section 4.1) and consisted of a series of comparison tests on a touch screen divided into two distinct areas. The user had to decide whether the stimuli perceived when flying over each zone differed. All tests were performed with a single actuator. The user had to explore the tablet with the finger of one hand, and with the actuator placed on the other hand. The only question asked after each exploration with no time limit was whether the two stimuli on the left and right of the interface were identical. Initially, all amplitude values were set to a fixed value, and only the frequency values changed from one stimulus to the other. To minimize interference and maximize the fluidity of the experiment, a second tablet dedicated to the experimenter was connected to the first one by a Bluetooth connection. It was equipped with an interface to quickly and remotely control the presentation and the successive value of the series of stimuli. Each series consisted of a fixed reference value sent to the left side of the tablet and a variable comparison value sent to the right side.

The main result indicates that participants were collectively more sensitive to differences of frequencies close to 300 Hz. This perceptive ability becomes individualized and deteriorates as one moves away from this value. This result can be positively compared to [23], which indicates that tactile navigation on digital devices is more sensitive to vibration with frequencies within the range of 200–250 Hz. On the other hand, the experiment did not show any significant effect of amplitude noise, age or blindness (except for children who are significantly more sensitive to ascending than descending series). To summarize, the conclusions concerning the first grammatical rules applicable to our graphical language are declined according to the 5 frequency ranges listed below:Between 50 Hz and 150 Hz, the minimum perceptual threshold is 15 Hz.Between 150 Hz and 250 Hz, the minimum perceptual threshold is 13 Hz.Between 250 Hz and 350 Hz, the minimum perceptual threshold is 7 Hz.Between 350 Hz and 450 Hz, the minimum perceptual threshold is 10 Hz.Between 450 Hz and 550 Hz, the minimum perceptual threshold is 15 Hz.

These frequency values must be combined with amplitude values to increase the expressive power of the graphical language. Consequently, we designed a third experiment [24] to select the most perceptible amplitude range. Each studied amplitude value was further combined with the optimized frequency value obtained in the previous experiment. The protocol was very similar to the previous one excepted for the population (20 sighted adults and 5 blind adults (the same population as in Section 4.1), the reference values for amplitude ranged from 55 to 255 (Set to this range to be included in the communication frame and then translated by the micro-controller in amplitude value.), and the “staircase” and “up-and-down” method [25] used to determinate the amplitude perceptive threshold.

Many amplitude values were compared to three reference amplitudes by crossing the two populations of users. The best perceptive results are in an interval around the value 55 regardless of the experimental condition. Nevertheless, although measuring less sensitive perceptual thresholds, the higher values reveal a significant difference between the sighted and blind groups. Based on these results, we increased our grammar by the following 3 rules:Between 0 and 100, the minimum perceptual threshold is 12.Between 100 and 200, the minimum perceptual threshold is 48.Between 200 and 255, the minimum perceptual threshold is 45.

Through these experiments, we began to weave links between a graphical language and basic parameters of vibrotactile stimuli. In our last experiment, which is presented in the following section, we attempt to relate these results to the visual structures extracted from a corpus of 900 web pages.

## 5. Recognition of Web Page Structures

To test the perceptual capacities of the TactiNET through its graphical language, we propose an exploratory experiment based on the semi-automatic conversion of the global visual structures of web pages into vibrotactile stimuli. This conversion is achieved by dividing a given web page into meaningful clusters of bounding-boxes represented by rectangular shapes, and by translating them into vibrotactile stimuli. The final objective is to evaluate how these macro-structures can be perceived in a non-visual environment by using the graphical language of the TactiNET.

### 5.1. Protocol Design

To evaluate the perceptive capacities of our device, we propose to run an exploratory experiment, which tackles a small number of web pages from three different domains: tourism, e-commerce and news (these are the three domains that are tackled by the academic-industrial consortium of the TagThunder project funded by BPI France).

In particular, three representative web pages have been automatically chosen from a corpus of 900 web pages equally distributed by domain. The representative web page of a given category is the one that obtains the highest similarity score when compared to all the other web pages in the same set according to a similarity function proposed in [26]. The measure of similarity is based on the number of intersection points between the rectangular bounding-boxes that make up two pages when they are superimposed. Therefore, the web page that shows the maximum intersection points with all the web pages of the category is chosen as the most representative of the category. Based on this measure, the three representative web pages that have emerged from each category are www.francetourisme.fr (tourism), www.asdiscount.com (e-commerce), and www.fdlm.org (news) (see Figure 5).

After choosing the representative web pages, an agglomerative graph-based clustering algorithm described in [27] has been applied on each of them to obtain meaningful clusters representing their visual macro-structures. The algorithm starts with a web page and a given number of clusters to discover. Then, three phases are processed: (1) extracting the visual parameters of HTML elements (vision-based phase), (2) filtering and reorganizing HTML elements (DOM-based phase), and (3) clustering the extracted bounding-boxes (agglomerative graph-based phase).

For this experiment, we fixed the number of zones to 5. This number comes from [28], who showed that there is a limit in terms of the amount of information a person can receive, process, and remember. The theory, known as Miller’s law, proposes that the average number of objects one can hold in working memory is 7 +/− 2. Therefore, to guarantee that the number of chosen clusters does not affect the performance of the participants, we chose 5 clusters as the minimum information to be handled. This process is illustrated in Figure 6, where the rectangular shapes result from the agglomerative graph-based clustering algorithm applied to each representative web page. Please note that the clustering shows differences in surface, location, orientation, and spatial relations. The results from the clustering have then been adapted to our experimental setup to take into account the recommendations of [12] in terms of rendering forms for non-visual navigation on touchscreen devices. Figure 7 shows the adapted presentations of the vibrating pages representing each page of each category (the colors have no other interest than to clearly distinguish the areas for the experimenter). After adapting each representative web page, a particular vibrotactile feedback should be dedicated to each cluster. To exploit the light contrasts produced by the tablet, the relationship between the visual structure of the zones and the calculated vibrotactile stimulus is based on the standard deviation of the gray values of the pixels. The greater the visual contrast, the higher the standard variation value, the greater the strength of the tactile stimulus should be. Therefore, based on the results presented in the previous section in terms of perception thresholds, the standard deviation value calculated for each zone is proportionally associated with an amplitude value as mentioned in Table 2, while a constant value of 304.6875 Hz is assigned to frequency. A basic vibrating page was also constructed to serve as a reference by setting the amplitude to the optimum value of 55 for all zones.

By optimizing both the vibration feedback parameters and the difference in the generated visual structures, the experiment consists of making the hypotheses that (1) the TactiNET should increase the ability to discern different categories of vibrating pages (i.e., whether two pages shown are similar or different) and (2) better performance in this task should be achieved when using values of varying amplitudes. The experimental setup is illustrated in Figure 8. Eleven users participated in this experiment (5 sighted and 6 blind) as demonstrated in Table 3. The group of the 6 blind participants is balanced in terms of gender and the precocity of their blindness and their average age is around 54 years old. Five of the blind population regularly use personal computers but in different proportions (from 1 to 8 hours a day), and only one of them uses a smartphone to surf on the web. In parallel, all sighted participants are women with an average age of around 26 years old.

Overall, 36 web page structures comparisons have been performed by each participant. There are 3 comparisons of identical structures and 3 comparisons with different page structures. All comparisons are repeated 3 times to avoid random selection. In addition, all comparisons are subject to 2 amplitude conditions: variable or fixed. Under the same conditions as the previous experiment, each task included a series of tests, which present the compared structures on two touchscreen tablets. After the exploration phase, the participant had to decide whether both vibrotactile structures were identical or not. Finally, at the end of all 36 comparisons, the user had to draw the last discovered structure on a A4 sheet of paper.

### 5.2. Analysis of the Results

We are aware that the number of blind participants is not sufficient to perform statistically significant tests. However, gathering a large population of visually impaired people is (fortunately) not an easy task. Consequently, the analysis of this experiment will be exploratory and preparatory to determine specific hypotheses and to validate identified trends to be developed in a future larger protocol. Figure 9 gathers results for all the comparisons of all pairs of vibrotactile pages, representative of the following categories: news (N), e-commerce (E), and tourism (T). The scores are distributed in the four tables according to the 6 blind participants (D1 to D6), the 5 sighted participants (V1 to V5), and the condition of the amplitude value (set to 55 or associated with the contrast value of the original area set in Table 2).

Several general observations can be specified. First, the results are globally homogeneous between the two populations regardless of the amplitude condition with a ratio of about 60% correct answers overall (exactly between 61.1% and 64.8%). This score seems promising as (1) 5 out of 6 blind people never use tactile devices, (2) none of them had any training time, and the experience was long (on average 1.5 h) and boring, and (4) the device was in its minimal configuration i.e., one single vibrator and a maximum of one dimension of vibratory variation. However, we note a slight superiority of almost 4% for the blind population to correctly classify pairs with the variable amplitude. This remark is important as this superiority comes exclusively from the case of comparisons between similar structures. While the results in the four tables show that it is (logically) twice as easy to identify a dissimilarity between two vibrotactile structures as it is to be certain of their identity, the blind seem more able (by 10%) to exploit the variation in amplitude to remove their doubts about the similarity of two structures.

#### 5.2.1. Redrawing Task

Some remarks concern the quality of the drawings produced at the end of the experiment, as no prior information was given on the nature of the elements explored. The sighted people (eyes closed) were more comfortable in representing their perception naturally in the form of rectangles, but the drawing was often more complex than the structure explored (see Figure 10). In parallel, three of the non-sighted people could not abstract this notion and reproduced the path followed by their finger in the form of lines (see Figure 11).

However, two blind people reproduced the structures very faithfully, even though no formal indication had been given on the vibratory stimulus support (see Figure 12 and Figure 13). In any case, this exercise seemed to be an interesting task, even with a blind population of users, to evaluate the capacity of our stimuli to construct a mental representation of the visual structure. With a statistically more appropriate number of users, Figure 14 might show a correlation between the scores given to the drawings by experts and the comparison ones. This may be particularly true when extracting exclusively results from the blind subjects (Figure 14 right). In this case, the correlation appears to be more strongly linear than for the overall observation (Figure 14 left), where the V3 data may be an outlier.

#### 5.2.2. Navigation Strategies

As all the experiments were filmed, we were able to observe the different strategies used by the participants to explore the tablet touchscreen. A first typology is provided in Figure 15. What seems remarkable to us is the richness of the micro-strategies used and therefore the large number of possible combinations to anchor them in a global strategy (see Table 4). This experience does not yet allow us to define a clear relationship between combinations of micro-strategies and overall efficiency for the recognition of visual text structures. Nevertheless, the relationship with the pre-perceptual visual abilities seems obvious enough to extend this line of research, i.e., the aim is to lay a solid foundation for the exploitation of non-visual scanning processes and thus the development of effective tactile reading strategies. A preliminary analysis can be produced from the following three observations:

The majority of participants start navigating the tablets from the left to right. This might be due to the cultural habits related to the writing and reading directions.The participants use between two to six micro-strategies, which can be arranged into three classes: (1) continuous navigation taking information in both horizontal and vertical dimensions (NS1, NS2, NS3, NS10, NS11), (2) navigation using only one direction (horizontal, vertical or diagonal - NS4, NS6, NS7, NS8, NS9, NS12) and (3) navigation using none of these possibilities (NS5).The micro-strategy NS3 is used by all participants as well as NS4 to the exception of D5. It is also noticeable that there were remarkable differences between the participants in the time and the speed of using each micro-strategy.

These remarks lead us to make several hypotheses. As the structure being flown over is not known in advance, the natural strategies implemented are preferentially based on a continuous horizontal and vertical course of the entire screen. However, the efficiency of information capture is degraded by the sequence of too many or too different micro-strategies. These findings would participate to explain the three lowest scores of blind people: D1 (too many micro-strategies), and D5 and D6 (use of too many dimensions). The most effective strategies (D2, D3 and D4) have in common a two-step recognition. The information is first taken by a continuous path in both horizontal and vertical dimensions of the screen, and then followed by a verification, essentially vertical, to remove some uncertainties.

#### 5.2.3. Other Meaningful Results

Finally, looking at the detailed results by participant or by category of visual structures, some elements of discussion also emerge. First, only the “tourism” web page structure explored by blind people seems to influence the scores, i.e., scores are higher for blind people than for sighted participants. Indeed, by looking at the left-hand bar graphs (blind) of Figure 16 and Figure 17, the comparisons involving this visual structure are systematically (to one exception) superior in both the “similar” (yellow bar) and the “different” (brown and cyan bars) conditions. The explanation we propose is to be found in the congruence between the shapes flown over and the amplitude value chosen. Indeed, the tourism category is the only one for which shapes close in surface and size are associated with values of close intensity (see Figure 7 and Table 2). This conclusion, if proven by further experiments on a vast population, is very interesting to support our approach because it legitimizes the use of objective criteria (variance in contrast), extracted from the source document and transformed in an analogical way, to build a coherent tactile landscape. In any case, this also requires us (in the future) to weight the amplitude value associated with the change in contrast of the zone by an area value of the zone.

Second, there seems to be a tendency for scores to be higher, the earlier the onset of blindness is, as shown in Table 5. In addition to the conditions for the occurrence of blindness, we will note in our results a tendency towards an easier task for blind people who are familiar with new tactile technologies, and for those who spend a lot of time surfing the web. In fact, the best performance is attributable to the only blind woman in the protocol (i.e., D3 with nearly 90% correct answers under all conditions), who is the youngest, has an iPhone equipped with VoiceOver, and is connected for more than 10 h a day.

## 6. Conclusions and Perspectives

In this paper, we developed a vibrotactile framework called TactiNET for the active exploration of the layout and the typography of web pages in a non-visual environment, being the idea to access the morpho-dispositionnal semantics of the message conveyed on the web.

For that purpose, we first built an experimental device allowing the analogical transposition of the light contrasts emitted by a touch-sensitive tablet into vibratory and thermal stimuli. The main ergonomic constraints were to be easily positionable on any part of the body, modular in terms of the quality and the quantity of actuators, inexpensive (less than 100 dollars), robust and light.

We then tuned the device to lay the bricks of a tactile language based on the expressive capacity of the stimuli produced. This work was initiated by the study of the minimum thresholds of perception of the frequency and the amplitude of the vibration.

Finally, we evaluated the ability of the TactiNET to allow the correct categorization of web pages of three domains, namely tourism, e-commerce and news presented through a vibrotactile adaptation of their visual structure. Although exploratory, the experiments are particularly encouraging reinforced by the fact that we deliberately chose very unfavorable conditions, i.e., (1) heterogeneity of the blind population in terms of age, habituation to tactile technologies and web browsing, onset of blindness, and (2) minimal configuration of our device with only one vibrotactile actuator and one maximum dimension of variation. Despite this, the interesting hypotheses that we retain are:blind people tend to affirm the similarity between two structures better than sighted people, especially when the relationships between the form overflown and the perceived intensities are consistent;blind people seem to be capable of imagining the shapes they have felt without any prior indication of the stimuli;users seem to develop a rich set of micro-strategies to browse the vibrating touch screen;the regular use of touch technologies and the number of daily hours spent on the web seem to be a positive factor for the appropriation of the device.

This first exploratory study opens up many avenues of research. For example, one possible direction is to refine on pressure, spatial and temporal criteria the typology of micro-strategies for exploring forms. The objective will be to analyze the relationships between micro-strategies and macro-strategies to access to information. We believe that through this interactive alternation between local and global perceptions, our natural scanning and skimming capabilities can be exploited, whether in the visual or the tactile modality.

One of the main perspectives of this work is also to experimentally explore more complex configurations of our device and thus improve the expressiveness of our tactile language. First, we will propose to study the association of visual and thermal parameters. Second, work is in progress to evaluate the possibility of combining tactile stimuli with audio stimuli [29]. Third, increasing the number of concurrent stimuli (up to 320) by associating a vibrotactile actuator to several fingers may enable new sensory experiences as the touch screen might be accessed by a multi-sensory device.

Finally, we will note the emergence in our experience and by informal discussions with blind people of a possible extension of the language. Indeed, one of the frequent difficulties encountered by blind people is the difficulty of having an *a priori* idea of the size of a document. In our experiments, some of the shapes chosen to calibrate the device had a black to white gradation over their entire surface (i.e., the higher the gray level, the stronger the vibration of the actuator). Therefore, the gradation produced a continuously decreasing vibration as the finger flew over it. Many blind people reported feeling differences in the speed of this decrease from the very beginning of their exploration of the shape. In fact, we constructed by chance a stimulus that allowed blind people to anticipate the size of an area. Therefore, it did not seem to be necessary for blind people to fly over more than a few centimeters to discriminate these differences and to know in advance the complete travel time of the shape. Consequently, we will propose to study the interest of a triple relationship (size of a zone, surface gradient and amplitude strength).

## Figures and Tables

**Figure 1 micromachines-11-00376-f001:**
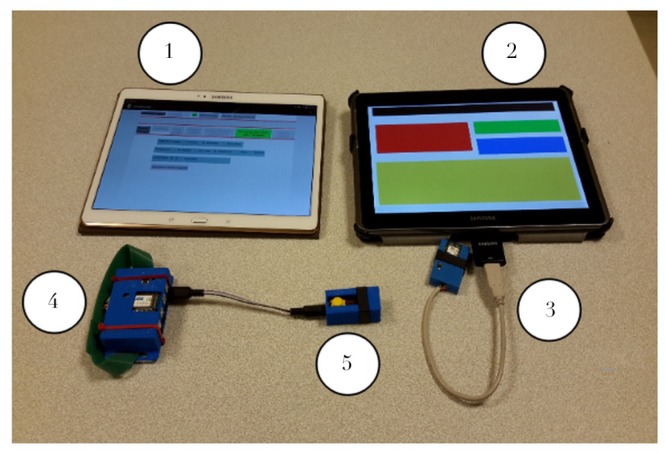
TactiNET: hardware setup.

**Figure 2 micromachines-11-00376-f002:**
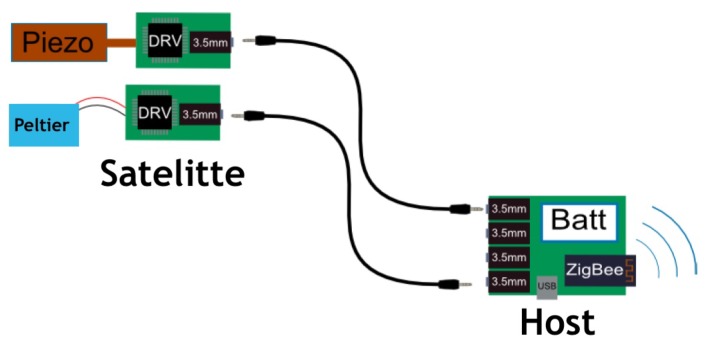
The host system communicates with the user’s tablet via ZigBee. Satellites are connected via 3.5 mm jack connectors and contain both piezoelectric or Peltier actuators.

**Figure 3 micromachines-11-00376-f003:**
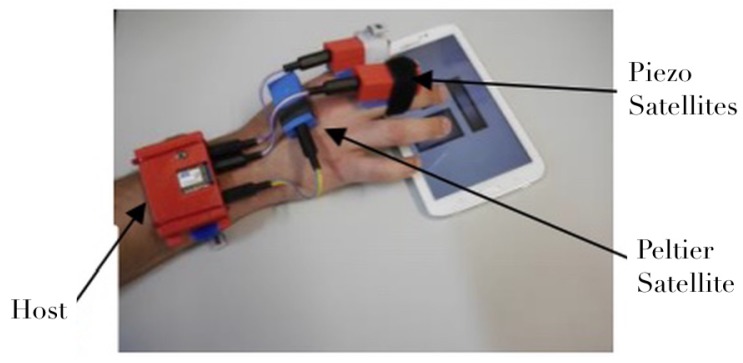
Example of assembly on active hand with one base, two piezoelectric and one Peltier.

**Figure 4 micromachines-11-00376-f004:**
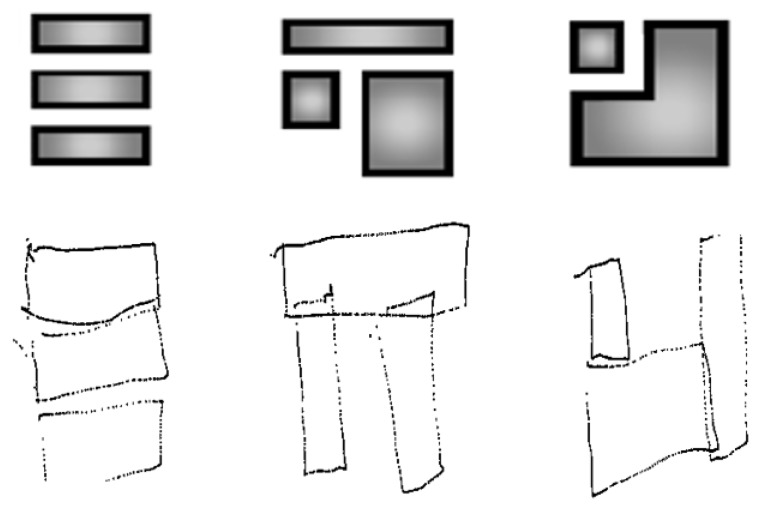
Original shapes and drawings of the perceived visual structures by the user.

**Figure 5 micromachines-11-00376-f005:**
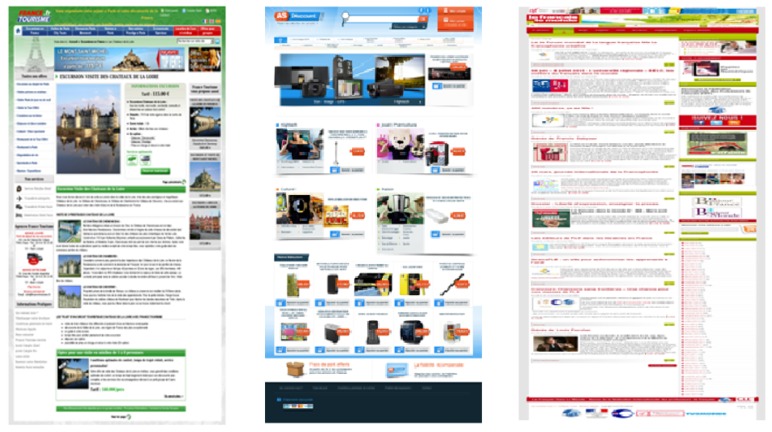
Three representative web pages: tourism (**left**), e-commerce (**middle**), and news (**right**).

**Figure 6 micromachines-11-00376-f006:**
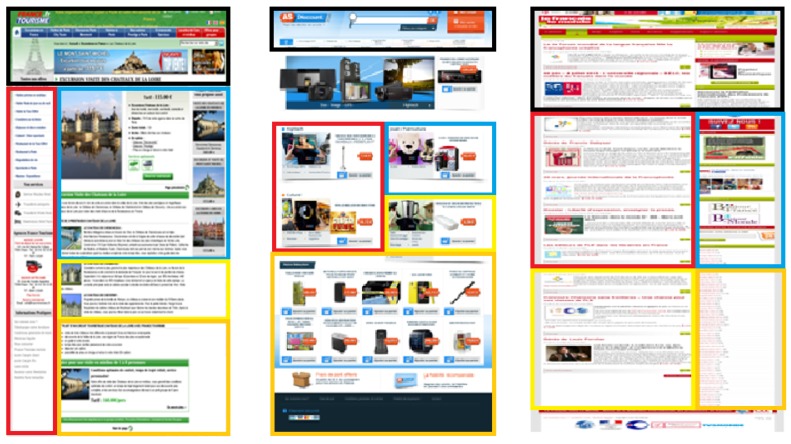
Clustering process of representative web pages.

**Figure 7 micromachines-11-00376-f007:**
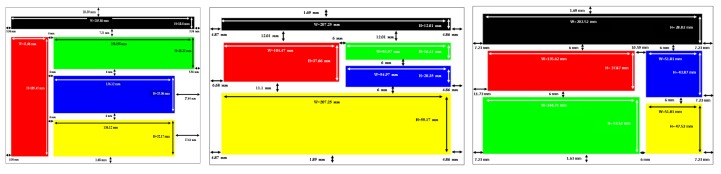
Adapted segmentation for experimental purposes.

**Figure 8 micromachines-11-00376-f008:**
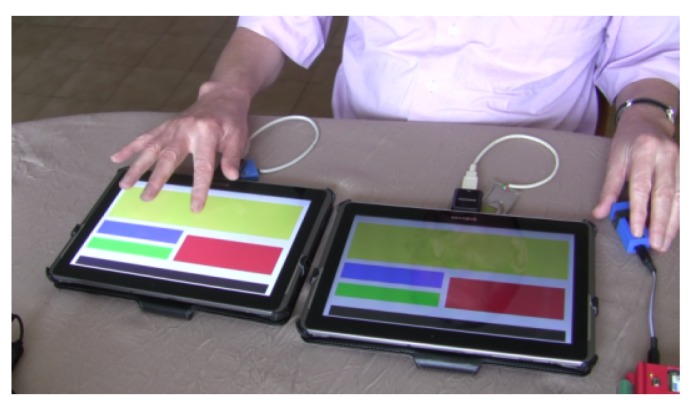
Comparing web page structures depending on associated amplitude values.

**Figure 9 micromachines-11-00376-f009:**
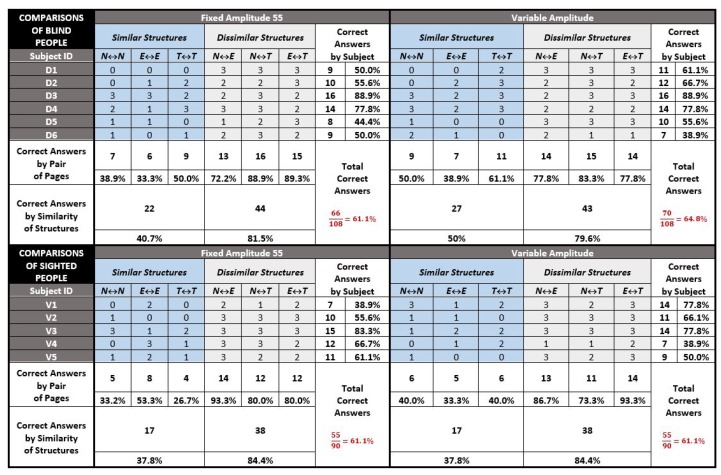
Results of all comparisons.

**Figure 10 micromachines-11-00376-f010:**
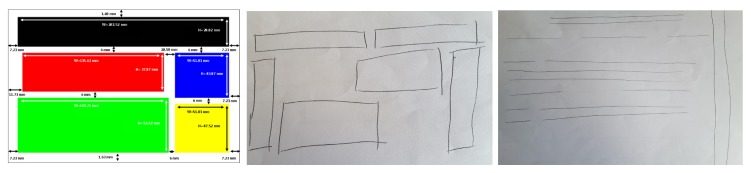
Sample drawings of V1 (middle) and V3 (right) in reference to the news structure.

**Figure 11 micromachines-11-00376-f011:**
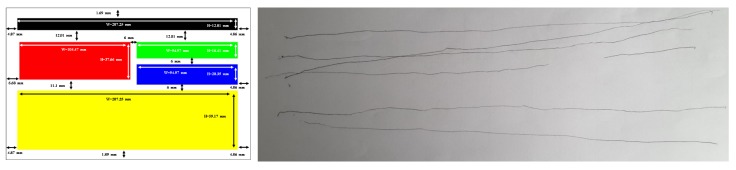
Sample Drawings of D4 in reference to the e-commerce structure.

**Figure 12 micromachines-11-00376-f012:**
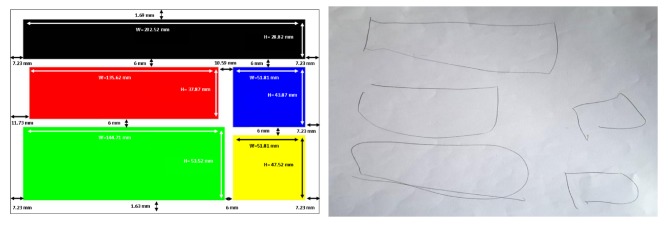
Sample drawings of D2 in reference to the news structure.

**Figure 13 micromachines-11-00376-f013:**
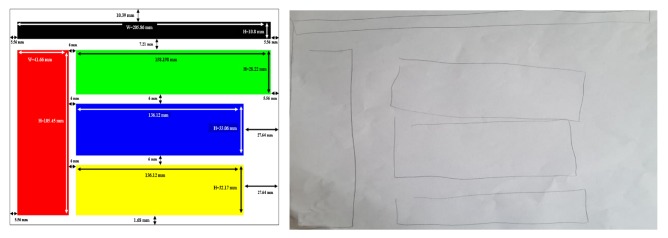
Sample drawings of D3 in reference to the tourism structure.

**Figure 14 micromachines-11-00376-f014:**
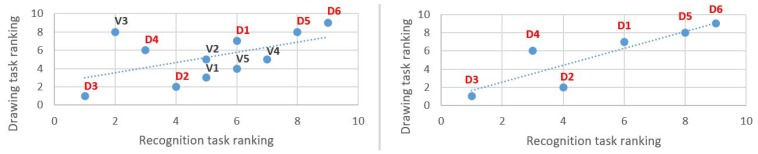
Recognition and drawing tasks: correlation for all subjects (**left**) and for the blind (**right**).

**Figure 15 micromachines-11-00376-f015:**
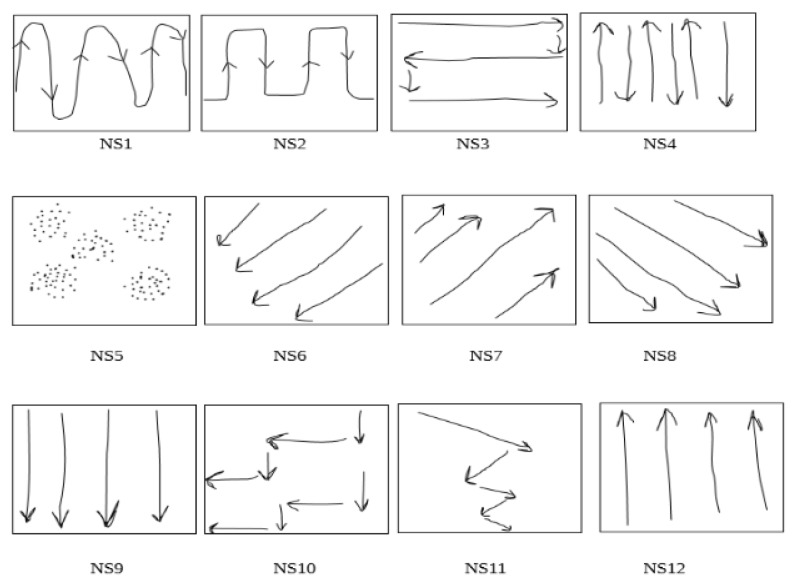
Navigation micro-strategies (NS) of blind and sighted participants.

**Figure 16 micromachines-11-00376-f016:**
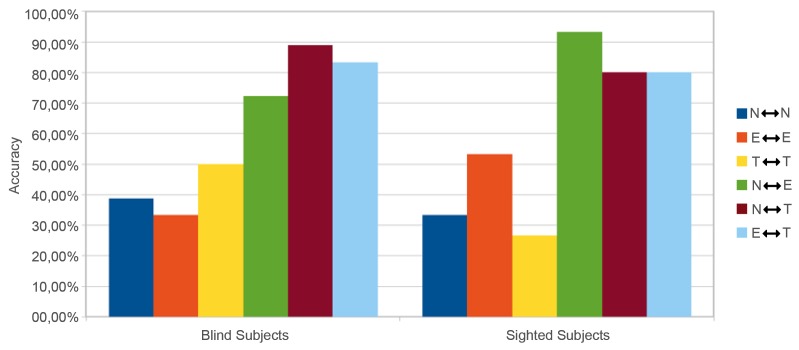
Accuracy by category - amplitude set at 55.

**Figure 17 micromachines-11-00376-f017:**
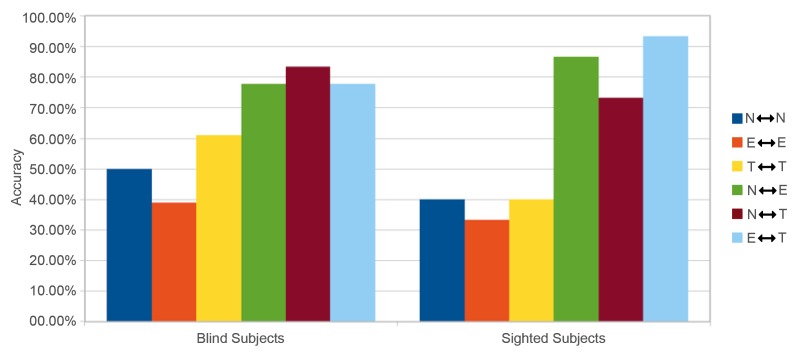
Accuracy by category - variable amplitude.

**Table 1 micromachines-11-00376-t001:** Characteristics of the blind population.

User id	1	2	3	4	5
Age	63	67	59	56	36
Sex	Male	Female	Male	Female	Female
Age of blindness	0	32	25	10	15
Operating system	Linux	Linux	Windows	-	Windows
Dedicated technology	ORCA	NVDA, ORCA	JAWS, NVDA	-	JAWS

**Table 2 micromachines-11-00376-t002:** Amplitude values for the vibrotactile stimuli.

Web Page	Black Zone	Red Zone	Green Zone	Blue Zone	Yellow Zone
**Tourism**	69	50	78	73	75
**Commerce**	79	80	72	65	95
**News**	69	57	52	77	31

**Table 3 micromachines-11-00376-t003:** Characteristics of the blind and sighted population.

User Id	Vision Status	Age	Sex	Age of Blindness
**D1**	blind	55	Female	0
**D2**	blind	66	Male	0
**D3**	blind	39	Female	8
**D4**	blind	40	Male	0
**D5**	blind	69	Female	32
**D6**	blind	60	Male	27
**V1**	sighted	27	Female	-
**V2**	sighted	26	Female	-
**V3**	sighted	25	Female	-
**V4**	sighted	29	Female	-
**V5**	sighted	25	Female	-

**Table 4 micromachines-11-00376-t004:** Navigation strategies (NS) chosen by each participant.

Subject ID	Chosen Navigation Strategies
**D1**	NS1, NS2, NS3, NS4, NS5, NS12
**D2**	NS3, NS4, NS12
**D3**	NS1, NS3, NS4, NS12
**D4**	NS3, NS4, NS10, NS12
**D5**	NS3, NS8, NS9
**D6**	NS3, NS4, NS8, NS9
**V1**	NS3, NS4, NS6, NS7, NS11, NS12
**V2**	NS3, NS4, NS12
**V3**	NS3, NS4
**V4**	NS3, NS4, NS12
**V5**	NS3, NS4

**Table 5 micromachines-11-00376-t005:** Congenital vision loss effect on accuracy.

	Fixed Amplitude		Variable Amplitude	
**Type of Participants**	**Similar Struct.**	**Dissimilar Struct.**	**Similar Struct.**	**Dissimilar struct.**
**All blind**	40.7%	81.5%	50.0%	79.6%
**Congenital visual loss**	50.0%	86.1%	63.9%	83.3%

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
