# Peer review of "Layout Transposition for Non-Visual Navigation of Web Pages by Tactile Feedback on Mobile Devices"

_micromachines, 2020, doi:10.3390/mi11040376_

Round 1

Reviewer 1 Report

The authors present TactiNET, a framework that helps blind people to navigate webpages on touch screen devices with multimodal actuator feedback on the hand, according to the screen content under the user's finger. The initiative seems novel and may play an important role in offering digital assistance to the people with impaired vision.

The approach is evaluated with 5 participants with normal vision and 6 blind participants.

The results support that the participants are able to figure out the correct webpage layout with the help of the system.

However, with only 11 participants, some claims shouldn't be drawn, for example line 493: 'there is a clear tendency for scores to be higher the earlier the onset of blindness' I don't see how such conclusion can be drawn with only 6 blind participants. How many participants have congenital vision loss? Without a large scale study, the individual differences in other aspects cannot be ruled out. the authors should explain the figures more explicitly in the caption for Figure 14 and 15. What are the y-axis?  The author should modify the writing with proper formal writing. For example: line 450: change 'whatever' to 'regardless of' It would be interesting to see the quantifiable time it takes the participants to navigate a page with the TactiNET, compared with traditional assistance methods

Author Response

1. We have added many cautions about the statistical value of our conclusions. In particular, this article is presented as a first draft of exploratory experiments whose objective is more to formulate credible hypotheses in order to produce scientifically valid results in the near future.

2. A table has been added to decribe participants profile

3. Captions and Figures have ben revised. Table readability has been improved.

4. we don't have the quantifiable time it takes the participants to navigate a page with the TactiNET, compared with traditional assistance methods 

5. To be more direct and clearer, the style of writing has been entirely corrected, the scientific questions highlighted, the sequence and size of the sections adapted to the subject matter.

Reviewer 2 Report

General Comments

I appreciate the efforts by the authors to develop a novel technique for blind people to interact with websites on mobile devices based on tactile stimulation. This is worthwhile research and tackles the complex and difficult aspects of sensory substitution or integration to help people with impairments their daily lives. However, the paper lacks a clear formulation of research questions and the discussion of the findings of the research.

The manuscript is generally too exhaustive in its descriptions. It could be substantially compressed and made more concise by cutting down the Introduction, Discussion and Conclusion parts to only those facts that are really relevant to the research presented here. There are several paragraphs that do not contribute much to the paper and that read more like a literature review or a narrative in a final research thesis than a research journal paper.

I am not sure if metaphors fit well into a scientific study. Personally I would prefer keeping the manuscript as concise and objective as possible.

Overall, I did not fully understand the research questions addressed by this paper and its findings. This should be improved by the authors together with presenting the results more clearly and in a more concise way.

Specific comments

Introduction

This part should be shortened substantially (by about 50%) to focus more on the most relevant previous research (instead of having an extra section on “related works”), the conclusions or limitations faced by the current state of the art and the research questions addressed by this research study. Please avoid narrative styles and keep the descriptions as tight and concise as possible to enable a better overview and clarity for the reader.

Part 3

The descriptions of the TactiNet hardware need more precise information about the tactile stimulation parameters. Instead of showing circuit board layout designs in Figures 1 and 2, you should concentrate more on the objective, physical descriptions and characteristics of the device used for the tactile presentation of website content. The layout of the circuit is arbitrary and does not provide much information on the perception or possible use cases of the device. Instead, what tactile perceptual aspects can the device satisfy and how did you choose/design the device to provide accurate and sufficient stimulation of the tactile system of the user?

For all of the design choices, please give reasons and explain why you chose this or that parameter and avoid saying that this can be configured as wished or changed by the user as this will complicate the replication of the research and its credibility.

Part 4.1 and 4.2

Are these descriptions of the previous research crucial for this study? How do they contribute exactly to the research study described in this paper? How does the current research build on these results? What are the parameters chosen based on these preliminary experiments and how are these incorporated into the design of the TactiNet device?

Part 5

This seems to be the main experiment part but I fail to understand the research question that motivated this experimental part.

The results are not clearly presented, as one fails to be able to make sense of the data given in Figure 10 (small fonts), Figure 11 (contrast), Figure 12 (shows no correlation, just a regression line - which may or may not be significant), Figure 13 (this is interesting - did people choose the same strategy for each trial? how many trials were there?), What is the difference between “Global” and “Specific” Results?, Figures 14 and 15 (no showing of variability in form of error bars, no label on the y-axis), overall the Figure captions are not sufficient to explain the data shown.

You should redesign this experiment to obtain data that is sufficient in terms of statistical power to perform statistical tests to answer the research questions you had. As mentioned above, I did not follow the research questions addressed here.

Discussion and Conclusion

These parts should be improved to relate the findings to other, previous research studies, to discuss potential limitations and further research to get a fuller picture of this research. The conclusion should state clearly and concisely the findings of this research study and allow the reader to quickly assess the findings - therefor the current manuscript does not fulfil this requirement as it is too exhaustive and not concise.

As inspiration, I would suggest looking for example at https://doi.org/10.3390/mi10100642 as baseline to improve this manuscript.

Author Response

A. General comments - Introduction - Part 4.1 and 4.2 - Part 5 - Conclusion

1. To be more direct and clearer, the style of writing has been entirely corrected, the scientific questions highlighted, the sequence and size of the sections adapted to the subject matter.

2. We have added many cautions about the statistical value of our analysis. In particular, this article is presented as a first draft of exploratory experiments whose objective is more to formulate credible hypotheses in order to produce scientifically valid results in the near future.

3. Table has been added to decribe participants profile

4. Captions and Figures have ben revised. Tables readability has been improved.

B. Part 3

It has been completely rewritten as requested and irrelevant diagrams removed. We made it clear that the design of a versatile tool was intended to provide flexibility for the experimenter (and not for the user) to test and compare the effectiveness of different configurations. These first results seem to us to be replicable and compared since all the experiments were carried out with the same configuration, precisely described.

Round 2

Reviewer 2 Report

I want to thank the authors for substantially revising their manuscript and for incorporating most of my suggestions. The manuscript, at least for me, is now much improved, the research questions are clearly addressed and the Introduction is much more concise and focused on the relevant literature and background for this research.

In particular, I appreciate the clear mentioning of the exploratory character of this research, the potential to provide the proposed system to other researchers working in this area and the honest discussion of the findings and their potential for future applications. 

There are further improvements to the MS, such as the Figures and descriptions are now helpful and clearer (they are still not perfect: Figures 14, 16 and 17 still have no axis labels and units, figure captions are very sparse still and the large table in Figure 9 could be presented better to convey the information to the reader). I leave this to the authors to improve upon.

It is interesting that subjects D1, D2 and D4 (blind from age 0) did not show any obvious differences in performance to D3, D5 and D5 (onset of blindness after vision). In line with this, there was no difference found between participants with or without vision. Instead, the authors described exposure to touch technology and/or the web/iphones as predictive of performance. This is promising and shows a clear potential to let people learn to use the TactiNet or similar devices irrespective of their visual abilities.

Also I find the last paragraph very interesting, and the "random" finding that the gradient was useful in predicting or estimating the overall size of the shapes during the exploring - this is also promising as a further dimension to convey visual information via tactile stimulation and I look forward to future research by the authors to explore this direction further.

Overall, I am convinced now that this study is a clear and appreciated contribution to this field of research and congratulate the authors on the improved manuscript.

Finally, there was only one typo that I found: "experiments" in the final paragraph.